

# Barnacles as biological flow indicators

Joseph W.N.L. Reustle[1], Benjamin A. Belgrad[2], Amberle McKee[2] and Delbert L. Smee[2,3]

[1] Department of Marine and Environmental Science, Hampton University, Hampton, VA, United States of America
[2] Dauphin Island Sea Lab, Dauphin Island, AL, United States of America
[3] School of Marine and Environmental Sciences, University of South Alabama, Mobile, AL, United States of America

Corresponding author
Joseph W.N.L. Reustle,
joereustle@gmail.com,
joseph.reustle@hamptonu.edu

## ABSTRACT

Hydrodynamic stress shapes the flora and fauna that exist in wave-swept environments, alters species interactions, and can become the primary community structuring agent. Yet, hydrodynamics can be difficult to quantify because instrumentation is expensive, some methods are unreliable, and accurately measuring spatial and temporal differences can be difficult. Here, we explored the utility of barnacles as potential biological flow-indicators. Barnacles, nearly ubiquitous within estuarine environments, have demonstrated notable phenotypic plasticity in the dimensions of their feeding appendages (cirri) and genitalia in response to flow. In high flow, barnacles have shorter, stockier cirri with shorter setae; in low flow, barnacles have longer, thinner cirri with longer setae. By measuring the relative differences in cirral dimensions, comparative differences in flow among locations can be quantified. We tested our hypothesis that ivory barnacles (*Amphibalanus eburneus*) could be useful biological flow indicators in two experiments. First, we performed reciprocal transplants of *A. eburneus* between wave protected and wave exposed areas to assess changes in morphology over 4 weeks as well as if changes dissipated when barnacles were relocated to a different wave habitat. Then, in a second study, we transplanted barnacles into low (<5 cm/s) and high flow (>25 cm/s) environments that were largely free of waves and shielded half of the transplanted barnacles to lessen flow speed. In both experiments, barnacles had significant differences in cirral morphologies across high and low flow sites. Transplanting barnacles revealed phenotypic changes occur within two weeks and can be reversed. Further, ameliorating flow within sites did not affect barnacle morphologies in low flow but had pronounced effects in high flow environments, suggesting that flow velocity was the primary driver of barnacle morphology in our experiment. These results highlight the utility of barnacles as cheap, accessible, and biologically relevant indicators of flow that can be useful for relative comparisons of flow differences among sites.

## INTRODUCTION

Hydrodynamic forces, such as flow velocity and turbulence can influence species interactions (*Hart & Finelli, 1999*; *Smee, Ferner & Weissburg, 2010*), structure communities (*Leonard et al., 1998*), and govern biodiversity (*Puijalon & Bornette, 2013*). Additionally,

the influence of hydrodynamic forces is clearly seen in morphology of intertidal flora and fauna (*Denny & Gaylord, 1996*; *Blanchette, 1997*) and in the foraging and predator avoidance behaviors of benthic organisms (*Menge & Sutherland, 1987*; *Smee, Ferner & Weissburg, 2010*). In wave-swept environments, organisms experience significant lift, drag, and acceleration and use both behavioral and morphological adaptations to withstand hydrodynamic forces (*Denny, 1985*). Behaviorally, organisms may change body orientation to minimize frontal area or adopt streamlined shapes, limit movement during intense flows, or seek refuge within microhabitats such as crevices (*Denny, 1985*, *Weissburg et al., 2003*). Morphologically, organisms may possess streamlined body shapes, higher area/basal strength ratios, or have different sizes and shapes of appendages than conspecifics or heterospecifics in low-flow environments (*Denny, 1985*).

Many marine organisms disperse broadly and experience wide variations in hydrodynamic conditions that may shift substantially over short periods of time or space. Slow moving and sessile organisms have adapted a variety of strategies in response to hydrodynamic conditions. For example, wave-swept algae typically feature flexible body plans that conform to streamlined shapes and limit applied forces and strong holdfasts that anchor them to the substrate (*Denny & Gaylord, 2002*). Sessile invertebrates alter both behavior and morphology to adequately respond to differences in hydrodynamic stress on short and long-term time scales. Behavioral plasticity is often the cheapest and most immediate tactic to limit exposure to deleterious conditions (*e.g.*, predators, hydrodynamics, temperature, *etc*). For example, barnacles will limit their feeding to lulls between waves to reduce exposure to mechanical stress (*Miller, 2007*). However, when unfavorable conditions persist, other costlier forms of plasticity may be necessary, often involving morphological differentiation (*DeWitt, Sih & Wilson, 1998*). Many marine gastropods grow a larger muscular foot without changing their projected surface area (surface area/basal strength ratio) in regions of high flow velocity to reduce the risk of dislodgement (*e.g.*, *Trussell, 1997*). Sessile organisms such as barnacles produce shorter, thicker feeding appendages in faster flows (*Marchinko, 2003*; *Marchinko & Palmer, 2003*).

Phenotypic plasticity in marine organisms may serve as bioindicators and provide a relatively inexpensive means to quantify relative hydrodynamic differences among locations. Methodological approaches to quantify hydrodynamics range from bulk flow investigations (*i.e.,* chalk-blocks, (sensu *Sanford et al., 1994*) to expensive instruments that make high precise measurements (*e.g.*, Acoustic Doppler Velocimeter (ADV) (sensu *Williams 3rd et al., 1987*). While the bulk-flow approaches are much cheaper and can easily be replicated, they only provide erosion rate data (mass lost/time deployed) and are sometimes questioned for their biological relevance (*Gaylord, 1999*; *Porter, Sanford & Suttles, 2000*). Further, chalk blocks are subjected to scouring, which can accelerate erosion rates in areas with strong waves or high sediment loads, limiting their usefulness in comparing broader spatial differences. In contrast, instruments that measure flow are often expensive (ADVs ∼$10,000–$20,000), making it challenging to accurately capture spatial flow differences (turbulence, wave periodicity, flow velocity, direction, *etc*.). Thus, using common marine organisms as biological flow meters can provide a useful mechanism to compare spatial differences in hydrodynamics (*Lunt, Reustle & Smee, 2017*).

Barnacles are a common fixture in coastal environments and are heavily influenced by hydrodynamics (*Trager, Hwang & Strickler, 1990*; *Leonard et al., 1998*; *Marchinko, 2003*; *Neufeld & Palmer, 2008*; *Pineda et al., 2010*). Hydrodynamics affect nearly every aspect of a barnacle's life cycle from larval recruitment, predation-risk, feeding behavior, morphology, and reproduction (*Arsenault, Marchinko & Palmer, 2001*; *Marchinko & Palmer, 2003*; *Marchinko, 2003*; *Neufeld & Palmer, 2008*). As sessile suspension feeders, balanid barnacles must be capable of coping with notable shifts in flow regime throughout their lifetime (*i.e.,* tidal shifts, seasonal shifts, storm-events, *etc*). Barnacles are phenotypically plastic and can alter the size and dimensions of their appendages (cirri and genitalia) (*Marchinko & Palmer, 2003*; *Marchinko, 2003*), as well as their feeding behavior in response to changes in flow (*Trager, Hwang & Strickler, 1990*). In areas of high flow, barnacles have predictably shorter, stockier cirri with shorter setae; in low flow, barnacles have longer, thinner cirri with longer setae. Behaviorally, barnacles respond to changes in flow direction instantaneously, even appearing to anticipate oscillating flow (*Trager, Hwang & Strickler, 1990*), while persistent shifts in flow illicit morphological responses that take place in only two to three weeks (or one to two molts) for some species (*e.g.,* *Balanus glandula* in *Marchinko, 2003*).

Several genera of barnacles are known to exhibit plasticity in their feeding appendages including *Balanaus*, *Chthamalus*, *Semibalanus*, and *Pollicipes* (*Marchinko & Palmer, 2003*). We performed two experiments to ascertain the utility of using balanid barnacles in the Gulf of Mexico (*Amphibalanus eburneus*) as flow indicators and to determine if plasticity in feeding appendages was related to wave action or flow velocity. First, we performed a reciprocal transplant experiment to measure effects of waves and current on barnacle morphology as well as if morphological changes could be reversed after a transplant. We also completed a second experiment in which we transplanted barnacles (*A. eburneus*) to low and high flow areas without waves and shielded half the barnacles to reduce flow speed to determine if morphological changes were related to to local flow conditions. Finally, we correlated these morphological changes to local Reynolds numbers to assess whether barnacles can serve as predictors of spatial differences in flow.

## MATERIALS & METHODS

The effects of waves and flow velocity were tested on barnacle morphology by performing a reciprocal transplant study at the mouth of St. Charles Bay near Goose Island State Park, Rockport, TX, USA (28.125062, −96.975864, Fig. S1, see accompanying video in supplementary files). Goose Island State Park allowed free access to leeward and windward areas for experiment 1. Previously, we found that the windward side of oyster reefs at this location experienced significantly higher waves that occur more frequently as well as higher flow rates and turbulence than the leeward side (*Lunt, Reustle & Smee, 2017*). Twenty PVC poles were deployed along these oyster reefs in April 2017 with ten poles on the windward side and another 10 on the leeward side and marked accordingly. After two months, the poles had noticeable barnacle recruitment, and we moved five poles from the leeward to the windward side of the reefs and vice versa to create four treatment conditions: (1) leeward (LW), (2) leeward then windward (LW/WW), (3) windward (WW), (4) windward

then leeward (WW/LW). Four weeks following the transplant, we recovered the PVC poles and returned them to the lab for processing. We measured 20 barnacles from LW, eight from LW/WW, 10 from WW, and nine from WW/LW using barnacles that were at least 1.0 cm apart to avoid confounding effects of crowding on morphology. After removal, the basal diameter of each barnacle was measured across the operculum, from carina to rostrum (sensu *Rasband, 1997*). Then, we dissected each barnacle and removed the 6th cirri and photographed it using an Amscope stereomicroscope with a mounted 14-megapixel camera. Photographs were uploaded into ImageJ and ramus length, width, and setae length were each measured using the segmented line function (sensu *Marchinko & Palmer, 2003*; *Lunt, Reustle & Smee, 2017*; Fig. 1). Specifically, ramus lengths were measured from the first segment after ramus bifurcation to the final ramus segment. Ramus widths were measured across the first basal segment. Setae lengths were measured from setae selected haphazardly from the middle endopodite of the cirrus. Basal diameters of barnacle sizes were compared among the four treatments (LW, LW/WW, WW, WW/LW) using one-way ANOVA. Cirral data were analyzed using ANCOVA with treatment as a fixed factor and basal diameter as a covariate to account for cirral characteristics correlating with barnacle basal diameter. Tukey post hoc tests were used for pairwise comparisons. Statistics were completed using JMP Pro 14.2.0 software.

Flow was measured on windward and leeward sides of oyster reefs using acoustic Doppler velocimeters (ADVs) with 2 ADVs placed on each side (Norktek USA™ vector model), and flow data from ADVs was analyzed using Explore V™ software. ADVs were deployed for 24 h, secured to measure flow 0.5 m above the substrate, and flow was measured at 8 Hz in 4-minute bursts every 15 min during the deployment. ADVs measure three-dimensional flow. The net flow velocity (U) was calculated by taking the velocity from each dimension $(x, y, z)$ and combining into a single value using the equation: $U = \sqrt{x^2 + y^2 + z^2}$. Flow velocity was calculated in this manner for each 4-minute burst and averaged across all bursts for each site. Turbulence was calculated using the root mean square (RMS) of the flow velocity data. Net turbulence was calculated, similarly to net U described above, for each 4-minute burst using the equation:

$$RMS = \sqrt{(RMS_x)^2 + (RMS_y)^2 + (RMS_z)^2}.$$

Flow characteristics are presented in Table 1.

We performed a second experiment in the northern Laguna Madre, TX, USA to measure effects of flow velocity on barnacle morphology in sites where waves were uncommon. Flow was measured at each location using NortekUSA™ ADVs and analyzed using ExploreV software as before (4-minute bursts at 8 Hz every 15 min). Three 24-h ADV deployments were performed over a 6-week period in summer 2018. ADVs were secured and measured flow at ∼0.5m below the water surface and at the same level as the transplanted barnacles which were always submerged (the area is microtidal). Net flow velocity and turbulence were measured and calculated using the procedures and equations previously described. Flow characteristics are presented in Table 2.

PVC poles were placed in the Laguna Madre ∼1 km from sites used for the planned experiment. After two weeks with noticeable barnacle recruitment, 10–15 cm segments of

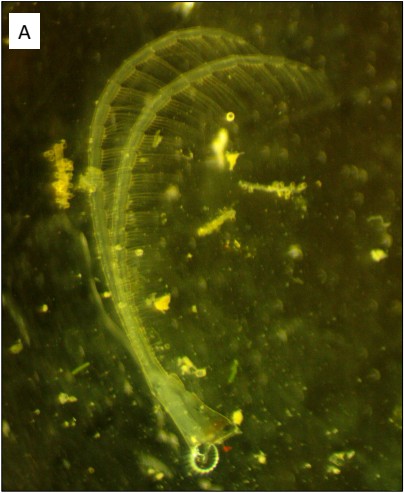 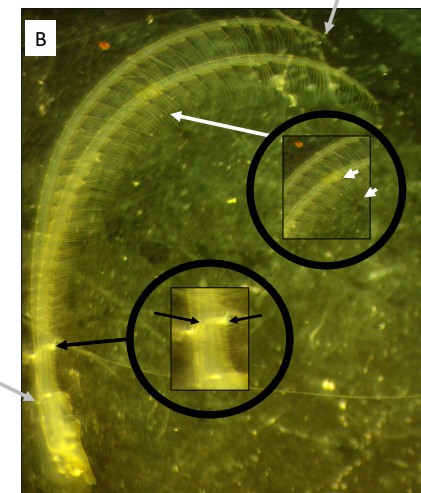

**Figure 1 Cirral trait measurement diagram.** (A) Windward and (B) leeward sites. Note the longer, thinner ramus present on the barnacle from the leeward side. Grey arrows highlight the starting and ending points of the curvilinear measurement of ramus lengths. Black arrows denote the locations for ramus width measurements. White arrows denote the measurement of the setae lengths from a segment within the middle endopodite.

**Table 1 Experiment 1: ADV measurements from the mouth of St. Charles Bay.** Mean flow velocity, turbulence, wave height, zero crossing period (average period for waves in a burst), and unidirectivity index (scale 0 −1 of wave direction with 1 indicating a single direction) at the mouth of St. Charles Bay.

| Location | Flow velocity (cm s$^{-1}$) | Turbulence (RMS) | Wave height (cm) | Zero crossing period (s) | Undirectivity index |
|---|---|---|---|---|---|
| Leeward | 3.75 | 3.4 | 5 | 1.3 | 0.95 |
| Windward | 6.68 | 18.6 | 22 | 1.5 | 0.98 |

**Table 2 Experiment 2: ADV measurements from high and low flow sites within the upper Laguna Madre.** Mean flow velocity and turbulence measured using ADVs at low and high flow sites in upper Laguna Madre. Average velocity (cm s −1) and turbulence (cm s −1) observed over three 24-hr measurements.

| | Velocity (cm s$^{-1}$) | | Turbulence (RMS) | |
|---|---|---|---|---|
| Date | Low-flow site | High-flow site | Low-flow site | High-flow site |
| 13-Jul | 3.55 | 16.92 | 3.74 | 15.73 |
| 3-Aug | 7.67 | 8.45 | 8.01 | 7.71 |
| 10-Aug | 3.36 | 59.96 | 4.25 | 20.99 |
| Overall Mean | **4.86** | **28.44** | **5.33** | **14.81** |

each PVC pole containing the greatest number of barnacles were selected for the experiment. After natural settlement, barnacles were far enough apart to prevent hummocking. Pairs of PVC segments containing barnacles were attached with zip ties to a PVC frame that

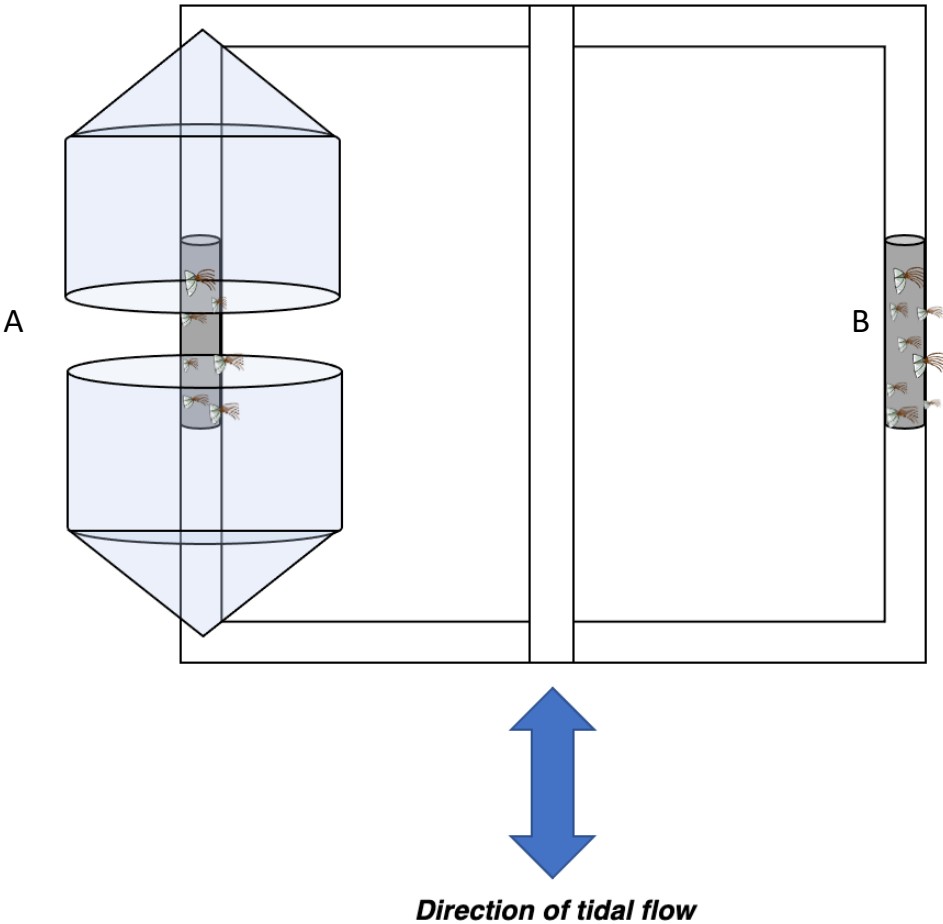

**Figure 2** **Barnacle shielding contraption diagram.** Schematic of barnacle shield deployed horizontally ∼0.5m below the surface in low and high flow areas. Conical plastic containers were used to modify flow regime within (A) protected treatments while (B) exposed barnacles were not shielded from flow in any regard.

allowed one of the PVC segments to be completely exposed to local flow conditions while the other PVC segment was protected from the local flow using a plastic shield (Fig. 2). Six frames, each at approximately 2 m between the next frame, were deployed into a low flow area (27.657315, −97.262032; with permission from the Cos Way Bait and Tackle) and six frames were deployed into a high flow area (27.621167, −97.215250; with permission from the LeCompte family) where they remained in the field from 7/01/2018 to 8/11/2018 (Fig. S2). Then, barnacles were recovered and taken to the lab for dissections.

We tested the effects of the plastic shield on flow by examining erosion rates of Life Savers™ placed behind the shield (protected) or beside the shield (unprotected). Erosion of Life Savers was calculated in a jar as a no-flow control and in field sites with different flow rates ranging from 5–25 cm s$^{-1}$ in close proximity to each other (<50 m apart total). Ambient flow conditions were measured using an ADV (salinity = ∼10 ppt, temperature = 28 °C). When protected from flow using the plastic shield, mass loss of Life Savers™ was

not significantly different from those in jars, while those unprotected lost significantly more mass than those in jars with mass loss increasing with flow velocity (Fig. S3), indicating that the plastic shield was effective in reducing flow.

We removed 51 barnacles from the low flow area and 28 from the high flow area, measured their basal diameter to the nearest 0.01 mm, and excised their sixth cirri. Dissected cirri were photographed using an Amscope SM-2TZ-LED-14M3 dissection scope. Cirral dimensions (ramus length, ramus width, and setae length) were then measured using the segmented-line tool in ImageJ version 1.51 (Sensu *Carlton, Newman & Pitombo, 2011*; *Lunt, Reustle & Smee, 2017*). The setae length of one barnacle was unable to be measured due to poor image resolution, but the other cirral traits were unaffected and were still measured for this individual.

Barnacle basal diameters were compared across locations and treatments using a two-way ANOVA with flow (high, low) and treatment (shielded, not shielded) as treatments. Each cirral dimension was analyzed using separate two-way ANCOVAs with flow level (low, high) and treatment (shielded, not shielded), and basal diameter as the covariate. Statistics were performed in JMP Pro 14.2.0. All barnacles (experiment 1 & 2) were collected under Texas Parks and Wildlife Scientific collection permit (SPR-0409-080).

### Reynolds number

Reynolds number is a proportional value describing the relative importance of inertial and viscous forces for movement relative to fluid (*Vogel, 1994*). Relatively large Reynolds numbers are indicative of stronger inertial forces, while small values are indicative of stronger viscous forces. For suspension feeding organisms like barnacles, Reynolds numbers may suggest whether the feeding apparatus is being dominated by inertial or viscous forces and functioning biomechanically like leaky sieves or like paddles, respectively (*e.g.*, *Cheer & Koehl, 1987*; *Geierman & Emlet, 2009*).

Reynolds number was calculated for barnacles using the equation:

$$Re = \frac{\rho \iota u}{\mu}.$$

Where $\rho$ is the density of seawater (1,025 kg/m$^3$), $\iota$ is ramus width, u is the mean flow velocity of the water recorded by the ADV, and $\mu$ is the dynamic fluid viscosity of seawater (0.0011 kg/m s). Only the unshielded barnacles were used for calculations of Reynolds numbers to examine ambient conditions. After Reynolds numbers were calculated, a Bartlett test on variance was calculated to determine homogeneity of variance. The Bartlett test indicated unequal variance between flow sites; therefore, Reynolds numbers were compared across flow regimes using a Welch's $t$-test in RStudio (version 1.1.456).

## RESULTS

### Reciprocal transplant experiment

In the first experiment comparing wave effects, leeward sides had smaller wave heights (LW: 0.05 m, WW: 0.22 m) and lower current speeds (LW: 3.75 cm s$^{-1}$, WW: 6.68 cm s$^{-1}$) as compared to windward areas (Table 1). Correspondingly, barnacles had larger

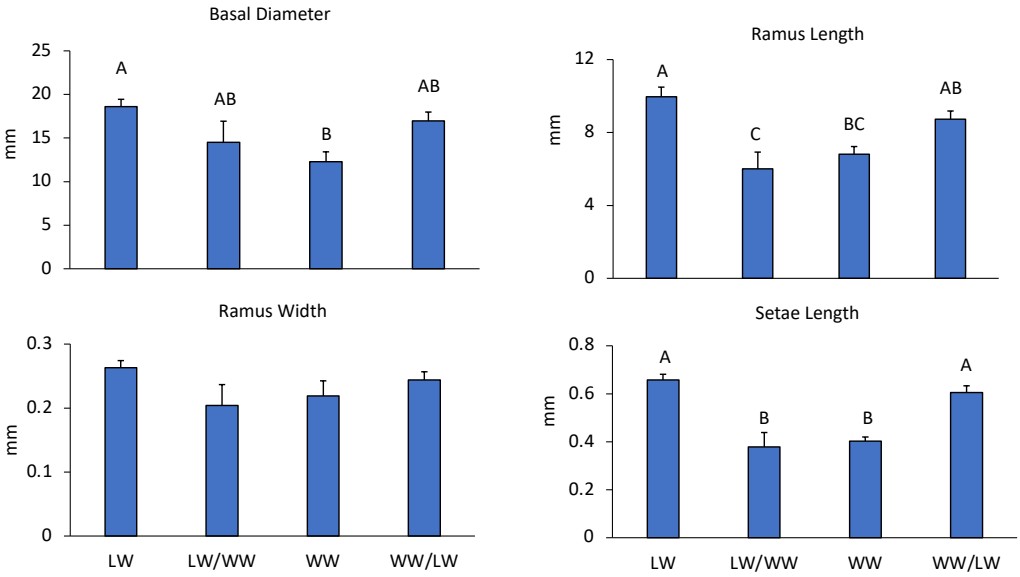

**Figure 3 Experiment 1: Reciprocal transplantations.** Mean + SE of four barnacle morphological measurements. Letters denote significant differences among treatments. LW = barnacles in leeward areas ($n = 20$), WW = barnacles in windward areas ($n = 10$), LW/WW barnacles from leeward areas moved to windward areas ($n = 8$), WW/LW barnacles from windward areas moved to leeward areas ($n = 9$).

basal diameters on the leeward side of the reef than on the windward side ($F_{3,43} = 5.41$, $p = 0.003$), while the basal diameters of transplanted barnacles were not significantly different from either leeward or windward sides (Fig. 3).

Barnacle cirri were significantly different between leeward and windward sites (Fig. 1). Barnacle ramus lengths differed between treatments ($F_{4,42} = 9.27$, $p < 0.001$). Ramus lengths were the longest for leeward and leeward transplants (barnacles transplanted from the windward side to the leeward side), while windward and windward transplants (barnacles transplanted from the leeward side to the windward side) had 20% shorter ramus lengths (Fig. 3). Leeward barnacle ramus lengths were not significantly different from leeward transplants, though they were significantly longer than both windward and windward transplants. Transplanted barnacles had significantly different ramus lengths (Fig. 3), while windward barnacles had intermediate ramus lengths that were not significantly different from either transplanted treatment. Ramus widths were not significantly different between treatments ($F_{4,42} = 1.53$, $p = 0.22$).

Setae lengths were significantly different between treatments ($F_{4,42} = 14.1$, $p < 0.001$). Leeward and leeward transplant barnacles had 16% longer setae than windward or windward transplant barnacles (Fig. 3). However, there were no intermediate lengths between treatments. Leeward and leeward transplants were not-significantly different from each other (Fig. 3). Similarly, windward and windward transplants did not differ significantly (Fig. 3). Essentially, wherever barnacles were collected from at the experiment end determined how they grouped by setae lengths, indicating that setae lengths change within two weeks.

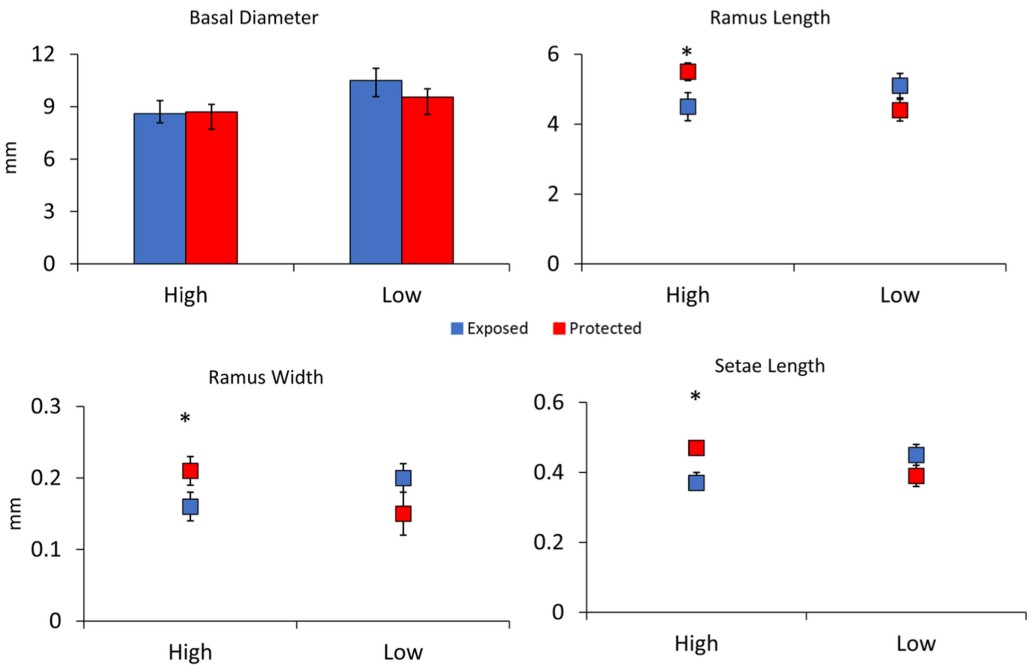

**Figure 4 Experiment 2: Barnacle shielding contraptions.** Mean + SE of four barnacle morphological measurements ($n = 40$ slow flow exposed, $n = 32$ slow flow protected, $n = 19$ fast flow exposed, and $n = 15$ fast flow protected). * indicates a significant pairwise difference between shielding treatments within sites. A significant interaction between site and treatment occurred for ramus length, ramus width, and setae length.

In the second experiment comparing barnacle morphologies in low and high flow areas without waves in the Laguna Madre, barnacles in low flow areas had 20% larger basal diameters as compared to those in high flow areas, and this difference was significant ($F_{3,78} = 2.30$, $p = 0.01$, Fig. 4). Differences between shield treatments for a particular flow regime were not significant ($F_{3,78} = 0.69$, $p = 0.49$, Fig. 4) nor was the interaction between flow and shielding ($F_{3,78} = 0.90$, $p = 0.36$, Fig. 4). Ramus lengths were significantly longer in slow flow areas by 12% ($F_{4,78} = 2.45$, $p = 0.02$, Fig. 4). Ramus lengths were not significantly different between shielding treatments (F$4,78 = 1.07$, $p = 0.28$, Fig. 4), but the interaction between flow and shield treatment was significant ($F_{4,78} = 2.43$, $p = 0.02$, Fig. 4) because the shielded treatment had large effects in the high but not the low flow site. Like ramus lengths, ramus widths followed a similar pattern being 20% wider in slow flow sites ($F_{4,78} = 2.06$, $p = 0.04$) with shielding treatment not significant ($F_{4,78} = 0.51$, $p = 0.61$) but a significant interaction between flow and shield treatment ($F_{4,78} = 4.09$, $p < 0.001$, Fig. 4). There was not a significant effect of flow ($F_{4,77} = 1.30$, $p = 0.26$) or shielding treatment ($F_{4,77} = 0.71$, $p = 0.40$) on setae lengths, but the interaction between flow and shielding was significant ($F_{4,78} = 6.14$, $p = 0.02$, Fig. 4).

### Reynolds number

Barnacles exhibited significantly different Reynolds numbers between high and low flow sites ($t = 10.2$, $p < 0.001$). On average, exposed barnacles in high flow sites experienced nearly 5x higher Reynolds numbers (high$_{Re}$ = 43.4, Low$_{Re}$ = 8.92).

## DISCUSSION

Cirri are the primary mechanism by which barnacles interact with their external environment, and the care and maintenance of cirri is necessary for survival. Thus, the apparent sensitivity of cirral dimensions to hydrodynamic conditions might be harnessed and incorporated into ecological studies as biologically relevant flow indicators. The notable plasticity of several species of Pacific balanid barnacles is well known (*e.g.*, *Marchinko & Palmer, 2003*), but similar comparisons of balanid barnacles from the Gulf of Mexico are limited. In this study, Gulf of Mexico ivory barnacles (*A. eburneus*) developed morphologies in response to the hydrodynamic conditions to which they were transplanted within four weeks when relocated between windward to leeward areas. This is consistent with earlier studies in Pacific balanid barnacles showing that both adults and juveniles can adjust their feeding appendage lengths in 18 days (*Marchinko, 2003*). Here, setae lengths changed first and on transplanted barnacles were indistinguishable from those that always grew in the transplant region. In contrast, basal diameters did not change among transplants and most barnacle rami did not converge to match natal barnacles. These differences in convergence rates across morphological features also suggest that differences in each feature can potentially account for differences in flow regime across different time scales (*i.e.,* setae lengths represent differences in flow occurring over ≤ 4 weeks' time whereas basal diameter represent overall flow differences occurring in time scales >4 weeks). Thus, by examining multiple morphological features simultaneously, researchers may gain insights into how flow regimes changed over time and across sites, although more research is necessary to pinpoint the time scales these features may represent. Interestingly, while the high flow regime experienced by windward barnacles was expected to cause individuals that spent their entire lives there to have the shortest ramus lengths of all treatments, the shortest ramus lengths were instead observed in barnacles that were transplanted to the windward side. This may be because the dramatic increase in flow may have caused transplanted barnacles to overcorrect in response to flow conditions. Alternatively, the sudden change in flow conditions may have rendered longer feeding appendages less effective, changing energy acquisition and growth parameters. Although we did not measure damaged or broken cirri, it is also possible that transplanted barnacles suffered damage to delicate feeding appendages that had an unsuitable morphology.

Localized differences in flow regimes created by shielding barnacles from water flow also produced distinct differences in barnacle morphologies in high flow sites. In low flow sites, ambient flow was not sufficiently different from flow blocked by the shield to trigger morphological changes in barnacle cirri. In contrast, in the high flow site, the shield had a significant effect due to large differences in flow between ambient and shielded conditions. These results in the high flow site indicated that *A. eburneus* barnacles can

exhibit local-scale phenotypic plasticity when hydrodynamic conditions differ on small spatial scales. Further, these differences between flow-exposed and protected barnacles suggest that plasticity was driven by flow and not by other environmental differences. Thus, local-scale hydrodynamic differences within a site may be quantified by measuring spatially clustered groups of barnacles while larger scale differences between flow regimes may be measured by averaging barnacle characteristics across multiple barnacle clusters within a site. Future studies are needed to determine the thresholds at which barnacles adjust their morphology. Incorporating other field sites with different abiotic conditions would also be useful to verify the conditions and thresholds under which this species exhibits plasticity in feeding appendages.

Barnacles experienced vastly different Reynolds numbers between high and low flow sites, which can affect the performance of suspension feeding appendages (*Cheer & Koehl, 1987*). Barnacles exposed to ambient flow conditions experienced approximately 5 times greater Reynolds numbers in high flow conditions than in low flow conditions; even the smallest barnacles in high flow experienced far greater Reynolds numbers than the largest barnacles in low flow conditions. At higher Reynolds numbers, cirral nets often become "leakier" and function more as rakes (or sieves) than paddles (at Re <1) (see *Cheer & Koehl, 1987*; *Geierman & Emlet, 2009*). Together, these data show that barnacles were both morphologically and functionally distinct across flow regimes.

The results from these experiments highlight both the significance of local-scale environmental conditions and the connections between form and function for suspension feeders. Previous works on barnacles have documented significant morphological plasticity in barnacle feeding appendages when transplanting barnacles from the open coast to protected harbors (*Marchinko, 2003*). However, as sessile invertebrates, barnacles experience significant environmental variation as tidal heights and magnitude change seasonally within the same site. Further, localized conditions change with the presence or absence of neighboring flora and fauna. For instance, within mussel beds, mussel-sheltered organisms may experience as much as a 30–62% reduction in wave energy as exposed organisms (*O'Donnell, 2008*). Such dynamic environments present considerable challenges for organisms, especially those which are sessile and cannot escape harsh conditions. *In-situ* manipulations of flow induces morphological modifications that balance the need to adequately capture prey with the risk of damaging key biological structures (*Marchinko, 2007*). Cirri are also key sites for gas exchange (*Resner et al., 2020*) and limiting damage to cirral structures is paramount.

Here, we document the utility of barnacles as biological flow indicators for assessing spatial differences in flow. We note differences in barnacle morphology in areas with and without waves, indicating barnacles can be useful flow indicators in a variety of hydrodynamic conditions. Careful consideration must be paid to morphological limitations in high flow (*e.g.*, >4 m/s as seen in *Li & Denny, 2004*), but also the size class of the barnacle influences the functionality of the cirral net, and the Reynolds number generated by the beating of the cirri (*Geierman & Emlet, 2009*). On one occasion at our high flow site, flow velocities exceeded those found by *Li & Denny (2004)* to cause cessation of feeding in barnacles. If flow velocities reduced barnacle feeding in our high flow site, this may explain

why barnacles had higher growth rates in low flow sites. We demonstrated great potential for the application of using barnacles as flow indicators in areas with and without waves. Given the incredible structural power of hydrodynamics in nearshore and shallow-water ecosystems, using barnacles as flow indicators represents a biologically relevant and accessible technique for investigations of recent and longer-term hydrodynamic conditions depending on which morphological characteristics are assessed.

## ACKNOWLEDGEMENTS

We are grateful for Smee and Dorgan lab members that assisted with the experiments. Goose Island State Park provided free access to leeward and windward areas for experiment 1. The Lecomptes family in Texas and Cos Way Bait and Tackle allowed us to use their docks for experiment 2.

### Funding
Funding for this work was provided by start-up funds from the Dauphin Island Sea Lab to Delbert L. Smee. The funders had no role in study design, data collection and analysis, decision to publish, or preparation of the manuscript.

### Grant Disclosures
The following grant information was disclosed by the authors:
Dauphin Island Sea Lab.

### Competing Interests
The authors declare there are no competing interests.

### Author Contributions

- Joseph W.N.L. Reustle conceived and designed the experiments, performed the experiments, analyzed the data, prepared figures and/or tables, authored or reviewed drafts of the article, and approved the final draft.
- Benjamin A. Belgrad conceived and designed the experiments, performed the experiments, analyzed the data, authored or reviewed drafts of the article, and approved the final draft.
- Amberle McKee conceived and designed the experiments, performed the experiments, analyzed the data, authored or reviewed drafts of the article, and approved the final draft.
- Delbert L. Smee conceived and designed the experiments, performed the experiments, analyzed the data, prepared figures and/or tables, authored or reviewed drafts of the article, and approved the final draft.

### Field Study Permissions
The following information was supplied relating to field study approvals (i.e., approving body and any reference numbers):

Goose Island State Park provided free access to leeward and windward areas for experiment 1. The LeCompte family of Texas and Cos Way Bait and Tackle allowed the use of their docks for experiment 2.

## Data Availability

The data from the barnacle transplant experiment, Life Saver erosion rates across contraption designs, barnacle data and ADV data are available in the Supplemental Files.

## Supplemental Information

Supplemental information for this article can be found online at http://dx.doi.org/10.7717/peerj.15018#supplemental-information.

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
