# Peer review of "Barnacles as biological flow indicators"

_PeerJ, doi:10.7717/peerj.15018_

## Round 0.1 · original submission · Minor Revisions

Both reviewers appreciate the work you've put into this paper, and have suggested some small revisions which will hopefully be straightforward to address

Reviewer 1 ·

Basic reporting

The manuscript is well-written and concise. The authors provided sufficient background information to provide important context about flow regimes and impacts on biota for the experiments. The figures are suitable to demonstrate the methods, which were helpful to visualize the approach, and also display the results. However, with Figure 1, I don't see any colored arrows. I see some red dots on panel B, but cannot distinguish any other colors or arrows on the panel. I don't know if this occurred during post-submission pdf formation. But please make sure these are clear in the final version.

Experimental design

The research question is well-defined and relevant. The methods are described in sufficient detail although I had a few questions.

1) on LN 128, it states that poles were deployed for 2 weeks and then the reciprocal transplants were completed. My understanding, then, is that these are early post-set barnacles. I am curious as to how this might affect the overall results - as these barnacles did not have much time to adapt to the initial conditions. For example, would a barnacle experiencing high turbulence conditions for 3 months change as rapidly as one that only experienced those conditions for 2 weeks?

2) on LN 131, it says the barnacles were collected four weeks following the transplant, yet at certain points in abstract and discussion makes it seem like they were collected only 2 weeks following the transplant (i.e. LN 37; 273)

3) LN 133, curious on the relatively low N here, less than 2 per pole for 3 of the 4 treatments. Also, please explain why barnacles that were at least 1cm apart were measured.

4) LN 177 - barnacles were spaced far enough apart to prevent hummocking. What does this mean? Initially, I interpreted these methods as using natural recruitment and selecting pieces of the PVC with the highest barnacle density, so how are they spaced apart? Were barnacles selectively removed to prevent neighbor effects? Please clarify.

5) LN 202 - not sure the t-tests are needed as the initial analyses (ANOVA and ANCOVA) already test within sites. Seems redundant.

Validity of the findings

The conclusions are well stated and are not overexaggerated. This was a relatively straightforward study, with clear results and the interpretation is sound. The authors do a nice job putting the results into the context of other studies of barnacles.

Reviewer 2 ·

Basic reporting

This is an interesting paper advancing the literature on the plasticity of barnacle morphology to flow in two manipulative experiments. The application of these results to using barnacle transplants or settlement as indicators of flow is relevant to the literature. I also really liked the combination of physics and biology to get at the question proposed. Something that was very interesting was that shielding in the low flow site did not induce any short-term differences, so maybe somewhere, the threshold or decoupling below when a change in flow will not be registered or distinguishable by the "biological indicator" might be a good addition to the discussion. I would have the stats checked by a statistician as well to be sure the correct approaches were used. For example, in experiment one, could a 2 way ANOVA be better (origin site and transplant site maybe?). Also because the differences seemed to be short-lived in some instances, there might be some addition to the discussion as to how long these barnacles reflect flow (for example, maybe it is only for the few weeks prior to collection?) and how different indicators have different temporal representation. The discussion regarding how barnacles on a PVC pole vs those in natural environments potentially shielded by other organisms or rocks and the nonlinear nature of flow in biogenic enviroments in the ocean might be expanded as this would be relevant to the application of barnacles as biological indicators of flow. Perhaps maybe only this technique would work by providing a settlement surface in each area, rather than by measuring existing barnacles? Overall I enjoyed reading the paper and my only concern was the overstating of the conclusion and as long as that can be toned down a bit, I think this contribution should be published. Some minor changes to consider follow. In the abstract, line 29, the hypothesis might want to be explicitly stated in the sentence "We tested...add hypothesis here... using". Seems like there is something missing from this sentence. Abstract line 40, I do think the conclusion is a bit overstated. Maybe use the words "most likely". Line 73, add period. Line 103, missing comma in citation. Line 110 change "was due to" to "was related with". Line 115 change "are the result of" to "were related with". In line 226, please check if you meant the greater than sign? I think you mean the less than sign? Also, when the p values are stated as less than 0.05, it would be useful the see the actual value, so maybe change all these statements to p = XX. Thanks. Line 267, change "can" to "might". Line 300 to 302, tone down the statement a bit, possibly using the word "suggests".

Experimental design

The experimental design is fine, though one wonders about the validity of using only one site to make these statements int he first experiment, for example. Although there is replication within the high flow and low flow sites (10 and 10), there is no replication of high flow or low flow sites. I run transplant experiments all the time and I know how difficult it is to maintain a transplant site and prevent vandalism and the like, but we do always try to replicate our sites as well as our treatment, even if the replictae site is nearby. In this case, because there were two experiments, though with different ends, the results are valid. Maybe replication of different flow regimes and sites might be a further recommendation in the discussion, as one always wonders if the results are somewhat specific to the site chosen. That could be tied into the threshold discussion mentioned above, where flow differences may not be detectable in the barnacles.

Validity of the findings

Findings are valid and as long as the conclusions are toned down a bit, I think this is an interesting contribution to the literature, as long as the stats are checked by a statistician.

---

## Round 0.2 · accepted · Accept

Thanks for addressing the reviewers' comments, and congratulations!